# Optimizing the Diagnosis and Biomarker Testing for Patients with Intrahepatic Cholangiocarcinoma: A Multidisciplinary Approach

**DOI:** 10.3390/cancers14020392

**Published:** 2022-01-13

**Authors:** May T. Cho, Sepideh Gholami, Dorina Gui, Sooraj L. Tejaswi, Ghaneh Fananapazir, Nadine Abi-Jaoudeh, Zeljka Jutric, Jason B. Samarasena, Xiaodong Li, Jennifer B. Valerin, Jacob Mercer, Farshid Dayyani

**Affiliations:** 1University of California Irvine Health, Orange, CA 92868, USA; nadine@hs.uci.edu (N.A.-J.); zjutric@hs.uci.edu (Z.J.); jsamaras@hs.uci.edu (J.B.S.); xiaodol14@hs.uci.edu (X.L.); jvalerin@hs.uci.edu (J.B.V.); fdayyani@hs.uci.edu (F.D.); 2University of California Davis Health, Sacramento, CA 95817, USA; sgholami@ucdavis.edu (S.G.); dgui@ucdavis.edu (D.G.); 3Sutter Health, Davis, CA 95616, USA; stejaswi@sutterhealth.org; 4Mayo Clinic Hospital, Phoenix, AZ 85054, USA; Fananapazir.ghaneh@mayo.edu; 5Helsinn Therapeutics (U.S.), Inc., Iselin, NJ 08830, USA; jacob.mercer@helsinn.com

**Keywords:** cholangiocarcinoma, biomarker testing, genomic alterations, precision medicine, next-generation sequencing, multidisciplinary, best practices, challenges

## Abstract

**Simple Summary:**

Cholangiocarcinoma (CCA) is a group of rare cancers that are associated with poor prognosis. Biomarker testing may provide CCA patients access to effective, personalized treatment options. However, several barriers exist to the seamless adoption of precision medicine in patients with CCA. Experts from the University of California (UC) Davis and UC Irvine identified gaps in standard practices for biomarker testing and recommended best practices, which are summarized in this review.

**Abstract:**

Cholangiocarcinoma (CCA) is a heterogenous group of malignancies originating in the biliary tree, and associated with poor prognosis. Until recently, treatment options have been limited to surgical resection, liver-directed therapies, and chemotherapy. Identification of actionable genomic alterations with biomarker testing has revolutionized the treatment paradigm for these patients. However, several challenges exist to the seamless adoption of precision medicine in patients with CCA, relating to a lack of awareness of the importance of biomarker testing, hurdles in tissue acquisition, and ineffective collaboration among the multidisciplinary team (MDT). To identify gaps in standard practices and define best practices, multidisciplinary hepatobiliary teams from the University of California (UC) Davis and UC Irvine were convened; discussions of the meeting, including optimal approaches to tissue acquisition for diagnosis and biomarker testing, communication among academic and community healthcare teams, and physician education regarding biomarker testing, are summarized in this review.

## 1. Introduction

Cholangiocarcinoma (CCA) is a diverse group of malignancies that originate at any point in the biliary tree [1]. Cholangiocarcinoma is classified as intrahepatic (iCCA), perihilar (pCCA), and distal (dCCA) based on the anatomical site of location, with each associated with unique presentations, biomarkers, diagnoses, and management (Figure 1) [1]. 

CCA has a poor prognosis with conventional therapies, with a 5-year survival rate between 7 and 20% [1]. Currently available standard treatment modalities include surgical resection, locoregional therapies, chemotherapy, and, more recently, targeted therapies [1,2,3]. While surgical resection or thermal ablation (intraoperatively, percutaneous, or endoscopic) are potentially curative treatments, only around 25 to 35% of patients with CCA are eligible for surgery [1,2,3]. However, recurrence after surgical resection is high and may occur in 50 to 70% of patients, underscoring the need for alternative approaches [4,5,6]. Liver transplantation may be an option in select patients with iCCA when complete resection cannot be achieved or in the presence of cirrhosis and tumors <2 cm [1].

Clinical studies support the benefit of chemotherapy in both first- and second-line settings for patients with advanced disease who are not amenable to surgical or locoregional therapies [1,2,3]. In the first-line setting, the ABC-02 trial demonstrated improved overall survival (median OS, 11.7 months vs. 8.1 months, HR 0.64, 95% CI 0.52–0.8; *p* < 0.001) and progression-free survival (PFS (median PFS, 8.0 months vs. 5.0 months, *p* < 0.001)) with cisplatin plus gemcitabine compared to gemcitabine alone [7]. Moreover, emerging evidence indicates promising results with first-line gemcitabine/cisplatin plus nab-paclitaxel in patients with biliary tract cancers; a Phase 3 trial evaluating this combination (NCT03768414) has completed accrual and results are awaited [8]. Following progression on cisplatin plus gemcitabine, the ABC-06 trial established FOLFOX as second-line therapy based on a modest OS survival benefit (median OS, 6.2 months vs. 5.3 months) using active symptom control (HR, 0.69, 95% CI 0.50–0.97, *p* = 0.031) [9]. 

To improve treatment options for patients with CCA, additional research has been conducted over the past decade to identify a molecular map for the development of novel, targeted therapies, which has led to recent FDA approvals of three targeted therapies in the second-line setting: pemigatinib, infigratinib, and ivosidenib [10,11,12,13]. Biomarker testing has revolutionized the treatment paradigm for patients with CCA. However, there are potential barriers to the implementation of precision medicine in patients with CCA. The barriers may include delays in CCA diagnosis, hurdles associated with tissue acquisition for biomarker testing, ineffective collaboration between the hepatobiliary multidisciplinary team (MDT), and lack of awareness of the importance of biomarker testing [14,15,16,17,18]. The integral role of an effective MDT for the diagnosis and optimal treatment of patients with cancer, including CCA, is essential in both the community and academic settings; however, challenges associated with optimal patient care remain, which highlight the need for improvement [19,20,21,22,23]. In particular, patients with metastatic CCA progress rapidly; thus, improved diagnosis, biomarker testing, and treatment options are needed in this setting.

Recognizing these challenges, the UC Davis and UC Irvine multidisciplinary hepatobiliary teams convened to identify gaps in standard practices and to define best practices and recommendations for iCCA patient care; this review summarizes their discussions on iCCA diagnosis, optimal approaches to tissue acquisition for diagnosis and biomarker testing, communication among academic healthcare teams and their community colleagues, and the need to improve education regarding biomarker testing among academic and community physicians. The discussion is focused specifically on iCCA; the challenges with ‘extrahepatic’ CCA (consisting of pCCA and dCCA), while important, are not thoroughly discussed due to limited scope. Moreover, to avoid confusion and be in line with the adoption of consensus terms for testing in precision medicine, the term ‘biomarker testing’ will be used hereafter when referring to testing that is performed to identify genetic alterations [24].

## 2. The Value of Biomarker Testing in iCCA

The poor prognosis of iCCA has provided the impetus to identify novel targets and therapeutic agents [10]. Biomarker testing has identified several genomic aberrations that represent unique targets for therapeutic intervention in patients with iCCA [1,10,25,26]. Intriguingly, approximately 40–50% of patients with iCCA harbor actionable mutations and alterations, including fibroblast growth factor receptor (*FGFR*) fusions/rearrangements (10–16%), isocitrate dehydrogenase-1 (*IDH1*) mutations (10–20%), and *BRAF* mutations (<5%); in addition, neurotrophic tyrosine receptor kinase (*NTRK*) gene fusions (<5%) and high microsatellite instability (MSI-H)/tumor mutational burden high (TMB-H; <5%) have been identified as rare but potential targets for therapy [10,15,25,27] (Table 1).

Recent studies have shown that patients with iCCA derive clinical benefit from targeted therapies [28]. Specifically, iCCA patients with *FGFR2* fusions/rearrangements were treated with FGFR inhibitors (FGFRi) in recent clinical trials [29,30,31,32]. FGFRi treatment in previously treated patients with advanced iCCA containing *FGFR2* fusions/rearrangements resulted in response rates of 21–40%, disease control rates of ~80%, and mPFS of 7–9 months [29,30,31,32]. Based on the efficacy and safety results of multicenter, open-label, single-arm clinical trials, pemigatinib and infigratinib received accelerated approval by the US Food and Drug Administration (FDA) for the treatment of adults with previously treated, unresectable locally advanced, or metastatic CCA with an *FGFR2* fusion or other rearrangement, as detected by an FDA-approved test [11,12]. In addition, other investigational therapeutics that target *FGFR* fusions/rearrangements are currently being tested in first- and second-line settings for metastatic iCCA, underlining the significance of *FGFR* inhibitors for the treatment of patients with iCCA containing *FGFR2* fusions/rearrangements [33].

Additionally, the FDA recently approved ivosidenib (an *IDH1* inhibitor) for adult patients with previously treated, locally advanced, or metastatic cholangiocarcinoma with *IDH1* mutation, as detected by an FDA approved test [13]. The approval was based on the results of a randomized, multicenter, double-blind, placebo-controlled clinical trial that showed a statistically significant improvement in PFS with ivosidenib treatment compared with a matched placebo (mPFS, 2.7 vs. 1.4 months; *p* < 0.0001) [34]. Targeted therapies directed toward other alterations, such as BRAF, NTRK fusions, and MSI-H, have also shown promising efficacy and safety results in iCCA [28,35].

The promise of precision medicine compares favorably to current chemotherapy options, which have relatively low survival (mOS 5.5–8.6 months) and objective response rates (ORR, 5–14.8%) in the second-line setting [9,36,37]. Furthermore, CCA patients treated with chemotherapy in the first line often progress rapidly or might develop liver decompensation and may not be eligible for further therapy. Strikingly, it is estimated that only 15 to 25% of patients receive second-line chemotherapy, which ultimately depends on the patient’s performance status and organ function [9,36,38,39]. These analyses do not preclude chemotherapy utilization in first- or second-line settings. However, they highlight the importance of early and accurate biomarker testing so that patients may consider the most beneficial treatment options.

Recognizing the emerging role of precision medicine for the management of CCA, expert guidelines and working groups (National Comprehensive Cancer Network, NCCN; European Society for Medical Oncology, ESMO) recommend biomarker testing for patients with unresectable and metastatic CCA [40,41]. Overall, the presence of actionable genetic alterations, recent FDA approvals, and promising efficacy/safety of additional targeted agents in development show that comprehensive biomarker profiling in the iCCA population is paramount.

## 3. Identifying Potential Challenges and Solutions for Optimal Treatment of Patients with iCCA

Several potential barriers exist in the management of iCCA patients, which may limit patients from receiving the highest-quality care; these include challenges in diagnosis, biopsy collection for biomarker testing, communication among multidisciplinary team members, and physician education around new diagnostic techniques, biomarker tests, and treatments. In this section, barriers and potential solutions to improving the diagnosis and treatment of iCCA will be discussed.

### 3.1. Diagnosis

#### 3.1.1. Challenges

In clinical practice, iCCA is rarely diagnosed in early disease stages because patients are often asymptomatic or present with non-specific symptoms, with iCCA often found incidentally (20 to 30%) through imaging performed for other clinical reasons [1,42]. Optimized plans to predict the diagnosis of iCCA at an early stage (resectable) are critical to further improve patient care [43]. Strikingly, a recent study involving patients with CCA (*n* = 707) showed that the median duration from symptom onset to CCA diagnosis was 19 months (range, 1 to 241) and initial misdiagnosis occurred in 35% of patients (*n* = 247) [14]. 

When a liver mass is identified by imaging and iCCA is suspected, the recommended diagnostic algorithm includes liver function tests, analysis of serum biomarkers, additional imaging of the mass, tissue acquisition by an interventional endoscopist or an interventional radiologist, and histologic diagnosis by a pathologist [40].

While serum markers such as carbohydrate antigen (CA) 19-9, carcinoembryonic antigen (CEA), and alpha-fetoprotein (AFP) are assessed in patients suspected of having iCCA, they are not in themselves diagnostic [40,44].

Imaging plays a key role in the diagnosis of iCCA; however, imaging alone is not diagnostic, and no consensus exists on the imaging technique or diagnostic criteria [40,45,46]. While there are some imaging features that can suggest iCCA on contrast-enhanced CT and MRI (capsular retraction, rim-arterial phase hyperenhancement, and progressive enhancement on delayed images), ultimately, imaging cannot be definitive in distinguishing primary iCCA from its ‘morphological mimics’ [46,47,48].

Histologically, iCCA is characterized by well to moderately differentiated adenocarcinoma with varying degrees of desmoplasia [49,50]. Rare histological variants are also described, including a variant showing squamous and anastomosing trabecular components; a mucinous variant containing large tumoral glans lined by a mucinous epithelium and luminal mucin accumulation; a clear cell type showing clear cytoplasm; a sarcomatoid variant showing large sheets of spindled cells; or showing hepatoid morphology [49,50]. Given that these histological patterns can mimic other tumors, even with available tissue biopsy, the diagnosis of iCCA is considered among a differential that would also include atypical hepatocellular carcinoma, metastatic disease, and other rarer primary hepatic malignancies.

Additionally, immunohistochemical patterns that are typical of CCA include CK7+, CK19+, and CK20- [50]. However, this pattern is also seen in patients with metastatic lesions from the pancreas, stomach, and small bowel [51]. Due to the absence of specific diagnostic immunohistochemical biomarkers, iCCA is commonly regarded as a diagnosis of exclusion when a liver mass is found without primary cancer detection via PET-CT or upper/lower endoscopy [48,51]. Further research is needed to help identify definitive diagnostic immunohistochemical biomarkers of iCCA. Therefore, in clinical practice, pathology may not typically provide a definitive diagnosis of iCCA, instead indicating a tumor of pancreatobiliary origin, underscoring the importance for diagnosis by an MDT, where imaging and histopathology results can be combined (Focus Box 1).

Box 1Barriers to expedient iCCA diagnosis.
Symptoms may be vague, non-specific or asymptomaticAbsence of histologic specific markers of iCCARobust diagnostic tests are unavailable and histologic features and techniques (Albumin ISH) may be underutilized
Abbreviations: iCCA, intrahepatic cholangiocarcinoma; ISH, in situ hybridization.

#### 3.1.2. Best Practices

To establish the diagnosis of iCCA and distinguish it from other tumors, it is recommended that specific imaging observations are considered and analyzed [46,52]. HCC may be distinguished from iCCA based on radiological features on a contrast-enhanced imaging study; while HCC lesions show hyperenhancement in the arterial phase and contrast washout in the venous phase, iCCA shows progressive contrast uptake throughout both arterial and venous phases without a washout pattern [46,47,48]. iCCA may also be suspected when liver capsule retraction and biliary dilatation are observed in the vicinity of the intrahepatic lesion [46,47].

Importantly, albumin in situ hybridization assays may aid pathologists in the diagnosis of iCCA and should be considered in the workup when iCCA is suspected. Research has shown that non-hepatic primary cancers are largely negative for albumin and that positive albumin in cancer cells has high sensitivity for iCCA [52,53,54]. While the assays can differentiate iCCA from liver metastases, it should be noted that it may not be useful in differentiating iCCA from HCC or pancreatic acinar cell carcinoma lesions; therefore, HCC and pancreatic acinar-specific markers should be used to rule out these tumors. Additionally, a negative result should not be exclusionary for iCCA diagnosis as the sensitivity of the assay has ranged from 64 to 99% and large duct iCCA is commonly negative for albumin staining [52].

Other diagnostic clues include the cholangiolar pattern, which comprises well-formed ducts with angular profiles that mimic antler horns [52]. Although the cholangiolar pattern can be used to distinguish iCCA, it is not present in most cases [51,52]. Experts suggest that the simultaneous presence of the cholangiolar pattern and albumin (which are unrelated) can be thought of as virtually diagnostic of iCCA [52]. The combination of specific imaging characteristics with detailed pathologic analysis may further improve iCCA diagnosis (Focus Box 2).

Box 2Best practices to optimize iCCA diagnosis.
**Improve diagnostic tests**
Utilization of Albumin ISH to differentiate iCCA from liver metastasesIncorporation of pathology techniques that identify cholangiolar patternAwareness of imaging characteristics common to iCCA—for instance, capsular retraction
Abbreviations: iCCA, intrahepatic cholangiocarcinoma. ISH, in situ hybridization.

### 3.2. Biopsy Collection for Biomarker Testing

#### 3.2.1. Challenges

Recent analyses with hepatobiliary tumors, including iCCA, show that acquiring sufficient tissue for biomarker profiling remains challenging; NGS biomarker testing failure has been reported in nearly 30% of cases [15,16,55,56]. Lamarca et al. reported NGS failure in 26.8% of tissue samples, which may have been due to challenges in biopsy tissue collection [15]. In a separate study involving medical oncologists participating in the NCI’s National Survey of Precision Medicine in Cancer Treatment, nearly 60% of physicians described difficulty in obtaining an adequate biopsy sample as a key contributing factor for not ordering biomarker testing; the findings of this study are not specific to iCCA but apply broadly to biomarker testing [57]. Together, these studies indicate that there is a clear need to improve tissue acquisition for biomarker testing, especially for cancers such as iCCA.

In clinical practice, both ultrasound (US) and computed tomography (CT) guidance are used to visualize the target lesion, with the choice dictated by lesion accessibility and physician preference. The NCCN guidelines recommend ultrasound as the imaging modality to use in iCCA [40]. The use of ultrasound contrast agents has also shown promise in the identification of tumor margins for biopsy. With respect to biopsy technique, since endoscopic retrograde cholangiopancreatography (ERCP) is typically not applicable for iCCA, techniques such as percutaneous liver biopsies are often required to access tumors [58]. Biopsy specimens may be collected with percutaneous core needle biopsy (CNB) or fine needle aspiration (FNA); there is currently no consensus on the biopsy technique. While CNB may provide higher tissue yield and tissue architecture, FNA may provide better cellularity (but less tissue yield) and improved nucleic acid quality/sequencing metrics [58,59,60]. Of note, a strong rationale is emerging for the combined use of FNAs and CNBs to increase tumor cell content based on FNA’s ability to extract epithelial tumor cells from tumor stroma, ultimately increasing the tumor cell content [58,59,60,61,62].

Biopsy success may also be limited by the tumor content that is acquired from the target lesions during the procedure [15,28,35]. Desmoplastic tumors such as iCCA contain large amounts of fibrotic tissue and large stromal areas, which are major obstacles for obtaining an adequate tumor sample for genomic analysis [28,35,43,58,59,60,63]. Biomarker analysis specifications of tumor sample size, number of tumor cells, tumor fraction, and amount of DNA yield are typically defined by the NGS platforms used; however, it is important to note that biopsies from liver lesions may require a high tumor cell content since hepatocytes have higher DNA content than other somatic nuclei, which can potentially negatively impact biomarker test success rates [64]. Relatedly, most commercially available tests require tumor cell content greater than 20% [17]. Recent analysis shows that there are challenges with the collection of tumor cell content in patients with CCA. Specifically, post hoc analysis from 28 diverse US health systems showed that of 633 biliary tact samples, 19% had tumor cell content <20% [17]. Additionally, Lamarca et al. attributed NGS failure in tissue samples predominately to insufficient archival tumor content (<20%) [15]. Moreover, once the tissue is collected, factors affecting specimen quality and nucleic acid cell content must also be considered and include elapsed time before fixation, type of fixative used, and the fixation time [65].

Tissue yield and biopsy success also correlate with the type and gauge of needle; the choice varies with the lesion size, location and structure, amount of tissue needed, proposed imaging modality, as well as the institution [66]. Typically, 16–20 G needles are used for CNB and 20–25 G for FNA, and 3–5 cores are collected for CNB and 2–4 for FNA, with the length of cores typically ranging between 1 and 3 cm [58,59,61,66,67].

Safety is an important consideration when using invasive biopsy techniques. Major adverse events from percutaneous needle biopsies for biomarker analysis range from 0.5 to 1.2% [66]. Although the understanding of liver biopsy risks is currently incomplete, potential risks include bleeding, pneumothorax if the lesion is in the dome, and tract seeding—although the latter is rare, especially with use of a co-axial technique [58,68]. For patients with unresectable disease, the NCCN guidelines recommend liver biopsy only after determining transplant status [40].

Research shows that biomarker testing is essential to improve iCCA patient care [10,69]. However, the timing of tissue collection is not standardized [70]. Clear direction to determine if patients should undergo biopsy collection for diagnosis and biomarker testing simultaneously does not exist [10,70,71]. In addition, there is a lack of clarity on whether patients with resectable disease should undergo biomarker testing. However, experts have recently advocated that biomarker testing be conducted as soon as possible after diagnosis and that surgical teams save tissue that may be used for biomarker testing to benefit patients who may relapse [10] (Focus Box 3).

Box 3Barriers to iCCA biopsy/tissue collection.
Imaging modalities are not standardizedIdentifying candidates for liver transplantation (these patients are not candidates for biopsy) and concern for adverse events associated with biopsy collectionTissue acquisition method (choice of needle gauge, CNB vs. FNA) is not standardizedROSE is not available or utilizedTissue is exhausted due to core samples not being separated into separate cassettes for diagnosis and biomarker testingSub-optimal tumor cell content yield due to fibrotic tissue and high stromal content presenceLocation of tumor in liver can be difficult to access and imageFeedback is not provided between pathologist and interventional radiologist (no closed-loop system)
Abbreviations:CNB, core needle biopsy; FNA, fine needle aspiration; iCCA, intrahepatic cholangiocarcinoma; ROSE, rapid on-site evaluation.

#### 3.2.2. Best Practices and Recommendations

While expert working groups recommend biomarker testing for patients with CCA, well-delineated guidelines are needed to help improve tissue acquisition [61,66]. Guidance is required with respect to when to perform the biopsy, which lesion to biopsy, needle gauge to use, number of cores to collect, length of cores, and specimen handling information.

Additionally, when iCCA is suspected, communication among the hepatobiliary MDT is essential so that tissue can be preserved for biomarker testing [43,70]. Specifically, an important recommendation to conserve tumor tissue includes asking pathologists to split each core sample into two specimens so that one is reserved for histology/diagnosis and the other for biomarker analysis [58]. This will help to avoid exhaustion of tissue for iCCA diagnosis and will allow the pathologist to conduct biomarker testing upon confirmation of a patient’s histologic diagnosis [58,70].

Challenges associated with biopsy collection may not be related to technical issues alone but may also involve gaps in communication and coordination among the MDT that manages iCCA patients. As a result, use of a standardized biopsy requisition form that incorporates a prescreening scoring system that is modeled after the MD Anderson Cancer Center (MDACC) score may assist in specifying specimen procurement expectations [58,72]. Additionally, to increase the adequacy of tissue samples for NGS and provide immediate feedback on biopsy quality, rapid on-site evaluation (ROSE) during the biopsy is encouraged [73]. Lastly, it is essential that feedback regarding biopsy quality and biomarker testing success is provided from the pathologist to the radiologist or medical oncologist performing the procedure to help the radiologist to modify, improve, and standardize the techniques used to collect biopsy specimens [62,66] (Focus Box 4).

Box 4Best practices to optimize iCCA biopsy/tissue collection.
**Improve the quality of biopsy specimens for NGS**
Define clear standards for biopsy collection, including specimen parameters, biopsy type and technique, and preservation parameters (consider MDA scoring system)Balancing the biopsy benefits and risksEncourage ROSE and the inclusion of a cytologist during biopsy procedures to optimize tumor cell contentConsider collecting FNAs with core biopsies to improve tumor cell contentMacro/microdissection of samples by pathology group to enhance tumor cell contentWhen ordering NGS, provide rationale and goals of biopsy specimen for biomarker analysisImprove communication and feedback of tissue sample quality between the radiologist and pathologist (create a closed-loop system and consider incorporating ordering physician into discussions)
Abbreviations: iCCA, intrahepatic cholangiocarcinoma; FNA, fine-needle aspiration; MDA, MD Anderson; NGS, next-generation sequencing; ROSE, rapid on-site evaluation.

A representation of the various elements needed to optimize biomarker testing in patients with iCCA is depicted in Figure 2.

A potential solution to the challenges with tissue biopsy is liquid biopsy, which represents a minimally invasive and logistically less challenging alternative associated with a lower risk of complications compared to tissue biopsy samples. Moreover, the turnaround time for obtaining NGS results from liquid biopsies is shorter than from tissue samples, which may be critical in a rapidly progressing disease such as iCCA and important for patients that may be eligible for enrollment in biomarker-driven clinical trials [16]. Robust evidence supporting the universal use of circulating tumor DNA (ctDNA) for iCCA is currently limited, but recent analyses have highlighted its potential to identify actionable alterations in iCCA [74,75,76]. Moreover, the sensitivity of liquid biopsy tests to detect fusions/rearrangements, such as FGFR2 fusions, may be limited compared to tumor tissue [74,75,77,78]. Although the utilization of ctDNA is an attractive alternative to invasive procedures, its detection of biomarker alterations in iCCA needs further development. Despite its limitations, ctDNA analysis should be considered for iCCA patients that do not have tumors amenable to CNBs and to monitor resistance mechanisms to targeted therapies since, despite lower sensitivity, the specificity of ctDNA is acceptable. Future iCCA studies should include robust analysis of the concordance between ctDNA and tissue, along with enhanced technology to improve fusion detection. Until liquid biopsies are standardized and FDA-approved as companion diagnostic devices for iCCA genetic alterations and treatments, image-guided tissue biopsies for biomarker testing should remain the standard of care for all patients with iCCA.

## 4. Best Practices among Hepatobiliary Multidisciplinary Teams (MDTs) to Optimize iCCA Diagnosis and Treatment

Multidisciplinary cancer care is considered the ‘gold standard’ to improve cancer patient outcomes [79]. Given the invaluable role of MDT in improving patient outcomes, the NCCN and other experts endorse the use of an MDT for biliary tract cancers [40,43]. The primary goal of an MDT is to standardize treatment algorithms while optimizing individual patient care [19,79]. The MDT may encompass a combination of clinicians at several levels, and may include providers in community centers, cancer clinics, and academic institutions, with a range of expertise levels. Of note, inclusion of research-oriented team members in specialized cancer centers may help to improve patient outcomes by providing expertise in standard-of-care, cutting-edge therapies and encouraging enrollment in clinical trials [23,80,81].

A well-functioning MDT is characterized by several features. First, the team members should comprise all specialties appropriate for the cancer type. For iCCA, the hepatobiliary MDT generally includes gastroenterologists, radiologists (both diagnostic and interventional), pathologists, medical and surgical oncologists, and radiation oncologists, with each playing a critical role in patient care [19,20]. The team must be well-organized, efficient, have a good leadership structure, and be able to critically examine the quality and outcomes of MDT decisions [79]. The team must meet regularly to discuss cases, with regular attendance and a quorum required to ensure that all specialties are present at each meeting [19]. The hepatobiliary MDT must understand the importance of biopsy collection procedures for NGS. Standardized and well-documented techniques for the collection of specimens for NGS that allow risks and benefits to be balanced should all also be emphasized among team members. Most importantly, to optimize biomarker testing, it is crucial to have effective communication and coordination among the members of the hepatobiliary MDT (Figure 3). Effective communication will help to ensure the following: collection of adequate tumor cell content at diagnosis, tissue specimen preservation by separating core samples for diagnosis and biomarker testing, and feedback on biopsy quality and sequencing results. This might be achieved by specifically stating during a case discussion that NGS testing (ideally by tissue) is recommended.

However, in real-world clinical practice, the implementation of effective MDTs with precision medicine-focused communication may be difficult at both academic and community-based practices [21,82,83]. Available evidence indicates the clinical benefit of coordination between physicians in the community and academic settings for the treatment of patients with iCCA, [23,84], where improvements in diagnosis and treatment occur when a community physician seeks a second opinion from a specialized cancer clinic or academic center [20,85]. For example, single-day MTD liver clinics (MDLC) at an academic tertiary care cancer center that reviewed imaging and pathology results from outside providers led to a change in interpretations in 18 and 10% of cases, respectively [85]. Moreover, of patients with BTCs who presented for a second opinion, 31% of patients received new treatment recommendations compared to the one proposed in the community setting [20] (Focus Box 5).

Box 5Multidisciplinary team barriers.
**Barriers relating to multidisciplinary team**
Deficient communication among team regarding the complexity of iCCA diagnosis and new techniques to aid in diagnosisSiloed knowledge and education around biomarker testing treatment optionsNo standardization on when to order biomarker specific testsLack of molecular tumor boards and team analysis of biomarker test results to determine most effective treatment options for patients 
Abbreviations: iCCA, intrahepatic cholangiocarcinoma.

Moreover, a precision medicine program requires a multidisciplinary focus, along with molecular tumor boards composed of expert personnel, and skilled administrative support [19,21,84]. Standard operating procedures should be adopted to optimize biomarker testing. For example, a precision medicine program initiated at a large community-based hospital in California allowed oncologists and molecular pathologists to standardize all tumor biomarker testing by identifying a biomarker testing vendor and by using a central pathology laboratory for NGS ordering and tissue processing [86]. Additionally, reflex biomarker testing including rare tumor types was initiated, allowing a molecularly trained pathologist to order tests for selected solid tumors. Such updates to the precision medicine program resulted in an increase in biomarker testing from ~50 to ~95% of advanced lung cancers [86]. This program also helped to drive interest in biomarker testing, precision medicine discussions at molecular tumor boards, and optimized patient selection and referrals to clinical trial programs. In this context, physician education was thought to be key in establishing a comprehensive cancer biomarker program; the potential of physician education in improving iCCA care is discussed in a later section (Focus Box 6).

Box 6Multidisciplinary team best practices to optimize iCCA patient outcomes
**Increase communication and coordination among members of the multidisciplinary team**
Communication among radiologist, pathologist, and medical oncologist to collect biopsy samples for biomarker at diagnosisIdentify which physician or physicians are responsible for ordering NGSInclude clear leadership structureInitiate regular team meetings and molecular tumor boards to provide optimal analysis of biomarker test results
Abbreviations: iCCA, intrahepatic cholangiocarcinoma; NGS, next-generation sequencing.

## 5. Opportunities to Improve Knowledge and Utilization of Biomarker Testing

As previously discussed, a well-documented rate of actionable genomic aberrations (40–50%) occurs in iCCA, with several therapies targeting these aberrations [10,15,25]. This has prompted recognition of comprehensive genomic profiling as essential in the practice of precision medicine [10,15,43]. Due to the rapid advances in precision medicine for CCA, the NCCN and ESMO guidelines recommend the use of biomarker testing for advanced CCA [40,41].

Physician education should include information around the value of biomarker testing and, importantly, the nuances of biomarker testing platforms that provide a molecular guide to the patient’s tumor [18,70,86,87]. Prior to ordering a biomarker test, it is crucial for physicians to be aware of the specific genetic alterations that the biomarker testing panel covers, along with the advantages and disadvantages of each test, and how to best integrate the biomarker test results into the patient’s treatment plan [83]. Table 2 [88,89,90,91,92] outlines the various tests that can be considered, along with advantages/disadvantages of each. However, the large number of commercial tests and different reporting systems can make it difficult for clinicians to remain updated [83]. As a result, the creation of delineated guidelines and recommendations that define how to integrate biomarker testing into hepatobiliary multidisciplinary care is needed (Focus Box 7 and Box 8).

Box 7Lack of physician education and utilization of biomarker testing in patients with iCCA.
**Lack of physician education**
Restricted access to molecular tumor boards and inability to communicate with NCI-designated cancer centers to help optimize patient diagnosis, biomarker testing, and treatment optionsReduced awareness of the importance of biomarker testing to detect targetable alterationsLimited training on differences among biomarker tests—NGS (RNA vs. DNA, tissue vs. liquid) vs. FISH vs. RT-PCRLack of knowledge of approved targeted therapies for iCCA
Abbreviations: FISH, fluorescence in situ hybridization; iCCA, intrahepatic cholangiocarcinoma; NGS, next-generation sequencing; RT-PCR, real-time polymerase chain reaction.

Box 8Strategies to improve physician education and utilization of biomarker testing in patients with iCCA.
**Encourage education of physicians about biomarker testing and precision medicine treatments**
Enhance physician knowledge around the advantages/disadvantages of each biomarker test (NGS-RNA vs. DNA-based, tissue vs. liquid, etc.) and interpretation of test resultsCreation of delineated guidelines and recommendations that clearly define how to integrate biomarker testing into the hepatobiliary multidisciplinary care, ideally at diagnosisProvide information about the importance of NGS results for precision-based treatment and patient inclusion in clinical trials through educational activities (development of iCCA multidisciplinary focus groups, CMEs, educational grants, in-person and virtual meetings)Improve communication between academic radiologists and community-based radiologists around standardized biopsy guidelines for biomarker testingDevelop iCCA-specific multidisciplinary focus groups to advance knowledge around the importance of biopsies for personalized medicine/biomarker testingConsider implementing virtual molecular tumor boards in academic setting that incorporates community physicians (regional or statewide) to help with complex cases and analysis of biomarker test results
Abbreviations: CME, continuing medical education; iCCA, intrahepatic cholangiocarcinoma; NGS, next-generation sequencing.

## 6. Discussion

### 6.1. Expert Recommendations

We recommend that academic and community physicians are trained on the importance of NGS profiling in iCCA for precision-based treatment and patient inclusion in biomarker-driven clinical trials. The implementation of molecular tumor boards illustrates how to educate clinicians regarding the importance of biomarker testing [21,22,83,93]. Additionally, the application of precision medicine programs in community-based oncology practices is not only feasible but has been shown to improve survival and reduce healthcare costs [20,21,22,82,83,94]. Lastly, recent analysis shows the achievability of collaboration between academic and community oncologists to identify the most effective treatment options [83,84,95]. Notably, recent analysis showed that iCCA patients diagnosed and treated at both community and referral centers had better outcomes when compared to community centers alone (combined: mOS, 13 months (95% CI, 9–17 mos) vs. community: 4 months (95% CI, 3–5 mos)). The authors attributed this difference largely to the increased receipt of effective treatments when compared to patients treated solely at referral or community centers independently [23].

We also recommend the efficient dissemination of information related to recently approved targeted treatments, current clinical trials, and new techniques that improve biopsy collection. Although a variety of resources exist to help clinicians with optimized patient care and precision medicine, there is a clear need to educate all physicians [18,21,84,95]. For example, the Sarah Cannon Research Center Molecular Oncology Support Services (MOSS) initiative sought to educate community-based physicians and research staff, which led to increased clinical trial enrollment, biomarker testing, and appropriate treatment [83]. The MOSS program, which is largely virtual, adopted several measures, including the incorporation of clinic-specific molecular tumor boards (MTBs), a genospace web-based clinical trial matching platform, and a web-based MolecularHELP portal that addresses questions submitted by email or phone [83]. Using these platforms as part of routine care, the MOSS program provided proactive, prompt, real-time biomarker analysis. This ensured that biomarker test orders were reviewed by MTBs deployed at individual clinics as soon as the clinician received the results, followed by the evaluation of patient eligibility for biomarker-driven clinical trials.

Based on the challenges identified with biomarker testing and multidisciplinary care, in practice and within the literature, diagnostic hurdles, biomarker testing, and treatment barriers have been summarized in Focus Box 1, Box 3, Box 5 and Box 7. Additionally, recommendations for improving diagnosis, tissue collection for NGS, and treatment decisions are provided in Focus Box 2, Box 4, Box 6 and Box 8.

### 6.2. Future Directions

Emerging evidence reveals continuing limitations and challenges for biomarker testing in clinical practice despite the presence of multidisciplinary teams. This was illustrated by findings from a large community hospital’s experience, where, despite proven benefit with the precision medicine program, several hurdles remained, including reimbursement issues, correlating biomarker test results with clinical trial opportunities, interpretation of complex results, and incorporation of cell-free DNA/liquid biopsies [86]. Such observations highlight the need for continued efforts to address these challenges to optimize outcomes for patients with iCCA.

Recently, ASCO hosted ‘Precision Medicine: Expanding Opportunities’, an inaugural event in ASCO’s new State of Cancer Care in America (SOCCA) event series, which discussed opportunities and challenges in precision medicine and community-based models that successfully adopted precision oncology programs [21]. We recommend taking this one step further so that future events can be dedicated to the implementation of precision-based programs for specific tumor types, including iCCA. It is imperative that future meetings, guidelines, and recommendations clearly define how to integrate biomarker testing into hepatobiliary multidisciplinary care, ideally at initial diagnosis. Future discussion topics may include the preferred tissue acquisition method, number of cores needed, core needle biopsy size, optimal tissue processing, handling, and ways to improve communication among the hepatobiliary MDT.

## 7. Conclusions by the Expert Multidisciplinary Team

The clinical relevance of NGS biomarker testing in iCCA for the identification of actionable genomic aberrations and subsequent application of precision medicine is well-recognized. Despite unprecedented advancements in precision medicine and improvements in overall patient outcomes, several hurdles associated with iCCA patient care remain and include: (1) difficulty in iCCA diagnosis; (2) lack of standardization for imaging and biopsy techniques; (3) difficulty in acquiring and processing biopsy tissue for NGS; (4) the need to optimize multidisciplinary patient management; (5) lack of awareness of the importance of biomarker testing; (6) knowledge of the differences in biomarker testing platforms; (7) familiarity with biomarker-driven clinical trials; and (8) knowledge of available, FDA-approved targeted therapy options for iCCA. These barriers may prevent iCCA patients from receiving the most optimized care and must, therefore, be addressed.

Foremost, the implementation of multidisciplinary care in both academic and community settings is essential. Multidisciplinary care can help to facilitate the expedient diagnosis of iCCA, optimized biomarker profiling, and ultimately result in achieving the highest-quality patient care. To this end, improved communication and coordination among the members of the patient’s MDT, particularly between the community-level physicians and academic centers, is needed. In parallel, due to the constant updates to the iCCA treatment paradigm, it is crucial to continue to educate all physicians about biomarker testing, new therapies, and available clinical trials. Since 85% of cancer care occurs in the community [86], it is essential that both academic and community colleagues continue to convene to overcome challenges associated with patient care for iCCA.

## Figures and Tables

**Figure 1 cancers-14-00392-f001:**
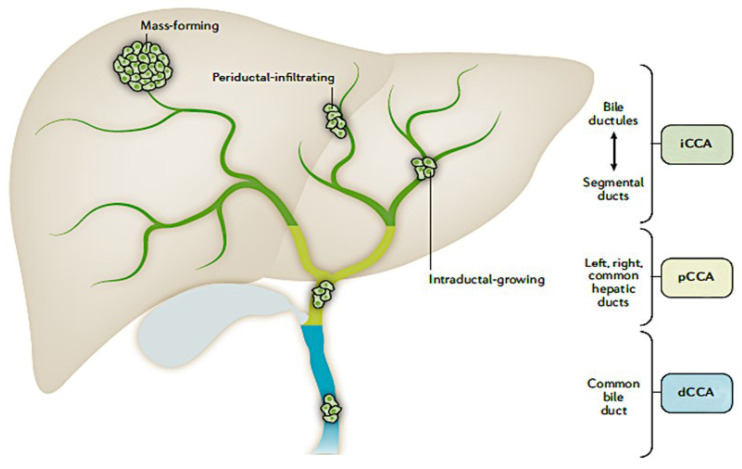
Location of cholangiocarcinoma within the biliary system and liver is reprinted from Banales JM et al. Nat Rev Gastroenterol Hepatol. 2020; 17: 557–588. Licensed under CC BY 4.0 (http://creativecommons.org/licenses/by/4.0/ Accessed on 8 November 2021).

**Figure 2 cancers-14-00392-f002:**
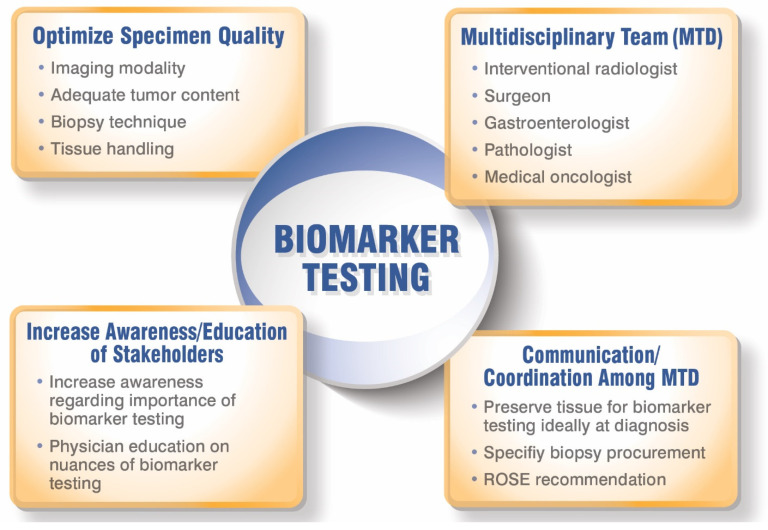
Optimizing biomarker testing for patients with iCCA.

**Figure 3 cancers-14-00392-f003:**
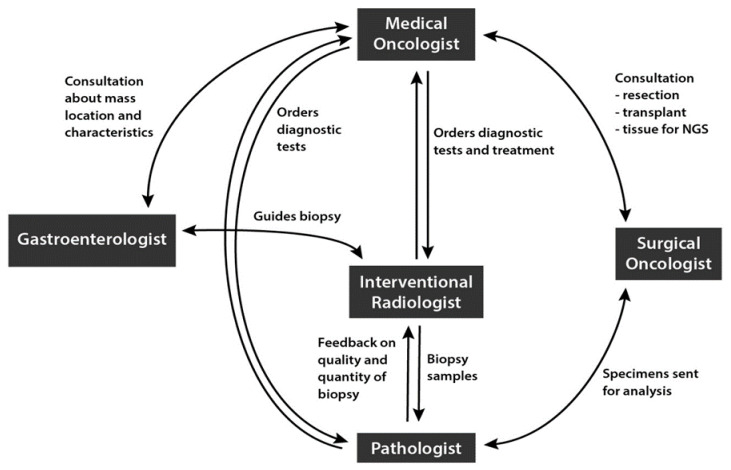
Communication and coordination within the multidisciplinary team in the diagnosis and treatment of iCCA.

**Table 1 cancers-14-00392-t001:** Actionable genetic aberrations in iCCA.

Gene	Prevalence *
*IDH1* mutations	10–20% [27]
*FGFR2* fusions/rearrangements	10–16% [25]
*BRAF* mutations	<5% [27]
*NTRK* fusions	<5% [27]
MSI-High/dMMR, TMB > 10 mutations/megabase	<5% [27]

Abbreviations: FGFR, fibroblast growth factor receptor; IDH1, isocitrate dehydrogenase-1; MSI, microsatellite instability; dMMR, deficient mismatch repair; TMB, tumor mutational burden. * The percentages provided are approximations.

**Table 2 cancers-14-00392-t002:** Biomarker testing options.

Test	Advantage	Disadvantage
Next-Generation Sequencing (NGS): Tissue-Based	Most clinically well-defined and studied test in CCA (FDA-approved companion diagnostics for FGFR2 fusions/rearrangements and IDH1 mutations) Comprehensive and generates largest amount of clinically actionable data, including fusions/rearrangements, point mutations, insertions/deletions/amplifications, TMB, MSI-H Detects multiple genes simultaneously and avoids exhaustion of tissue by single-gene testing panels	Longer turnaround times compared to single-gene panels, FISH, and liquid biopsy NGS Requires larger samples, with specific tumor cell content requirements Not all panels will be optimized to detect specific alterations (ex. fusions) - DNA-based tests may not be as efficient as detecting fusions/rearrangements when compared to RNA-based tests [88] Amplicon-based methods require prior knowledge of fusion partners and will miss novel fusions [89,90]
NGS—Liquid-Based	Noninvasive Shorter turnaround times when compared to tissue-based tests (NGS) Reduce risks of complications Valuable for serially monitoring resistance mutations during treatment and when biopsy/tissue is not available	Does not effectively detect certain alterations such as fusions/rearrangements Smaller gene panel sizes Limited analysis and data for patients with iCCA (ex. correlation of tissue and liquid tests are scarce) Dependent on higher tumor burden and may not be sensitive enough for early-stage tumors
FISH Testing	Detects genetic alterations, including fusions Less tissue is needed compared to NGS Shorter turnaround times when compared to NGS	Potentially exhausts tissue; not efficient or cost-effective when multiple genes need to be analyzed by NGS
Single-Analyte Tests (PCR/RT-PCR, Sanger sequencing)	Smaller samples can be utilized since limited genetic alterations are tested Shorter turnaround times when compared to NGS	Limited to known genetic alterations Potentially exhaust tissue—not efficient or cost-effective when multiple genes need to be analyzed by NGS, as is the case for iCCA
Immunohisto-chemistry (IHC)	Detects protein expression	Cannot detect genetic alterations such as fusions/rearrangements, point mutations, etc.

Abbreviations: FISH, fluorescence in situ hybridization; RT-PCR, real-time polymerase chain reaction.

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
