# Peer review of "Optimizing the Diagnosis and Biomarker Testing for Patients with Intrahepatic Cholangiocarcinoma: A Multidisciplinary Approach"

_cancers, 2022, doi:10.3390/cancers14020392_

Round 1

Reviewer 1 Report

In this narrative review entitled "Optimizing the Diagnosis and Biomarker Testing for Patients with Intrahepatic Cholangiocarcinoma: A Multidisciplinary Approach", Cho and colleagues reported the current body of knowledge concerning preoperative diagnosis and molecular profiling of iCCA.

Considering its aggressiveness, dismal prognosis and increased incidence worldwide, iCCA management must include early diagnosis and prompt molecular profiling for tailored chemo/immunotherapies

The paper is generally well written and organised, analysing several key issues concerning iCCA non-invasive diagnosis, sampling technique and recommendations for implementation of precision medicine program

The stylistic choice to highlight the current limitations in iCCA management and provide possible strategies for improve clinical practice is particularly intriguing and improves the paper readability 

I have a couple of suggestions:

1) I would suggest to switch the focus boxes into separate tables at the end of each paragraph in order to provide a more effective and easily accessible synthesis.

2) I would like the Authors to consider citing this review that has been recently published on Cancers, focusing on surgical approaches to CCA: https://doi.org/10.3390/cancers13153657

Best regards

Author Response

  1. We agree with reviewer comment. Per suggestion, we have organized the manuscript so that each paragraph with the specified challenge is followed by the focus box with the relevant points.
  2. Per reviewer suggestion, we have added the suggested citation [Lauterio et al. Cancers (Basel). 2021 Jul 21;13(15):365]

Reviewer 2 Report

This is a well-written and timely perspective on the best clinical management practices for a team treating a cholangiocarcinoma diagnosis. I really don't have much to add, though it would be nice if the authors could add just a few sentences on the specific challenges in metastatic patients.

Author Response

Per reviewer suggestion, we have made changes throughout the manuscript to highlight that the content is specific for metastatic patients. We have also referenced clinician's experience that metastatic cholangiocarcinoma patients often progress rapidly and thus improved diagnosis, biomarker testing and treatment options are needed.

Reviewer 3 Report

In this research, May Cho et al. provided a literature review of multidisciplinary approach to optimize diagnosis and biomarker detection in patients with intrahepatic cholangiocarcinoma. The review discussed potentially difficult challenges and solutions in determining the best treatment for patients with intrahepatic cholangiocarcinoma, both in terms of diagnosis and biopsy collection for biomarker testing. The review also discussed the importance of multidisciplinary approach in the diagnosis and treatment of intrahepatic cholangiocarcinoma. To best of our knowledgeable, there is no similar literature in recent years. Nevertheless, there are some issues for the authors to consider and solve.

  1. It was found that the content of Figure 1 in this review was the same as figure 1 in another review (PMID:32606456). Although the author cited this paper in the review, Figure 1 did not indicate that the figure was cited.
  2. In the second part of the review, the value of biomarker detection in cholangiocarcinoma was described in the article as cholangiocarcinoma, but table 1 shows actionable genetic aberrations in intrahepatic cholangiocarcinoma. So is this part of the review about cholangiocarcinoma or intrahepatic cholangiocarcinoma?
  3. In Table 1, when displaying the molecular characteristics of intrahepatic cholangiocarcinoma, some genes such as BAP1, which is also a common mutation in intrahepatic cholangiocarcinoma gene mutations, are missing.
  4. On page 10 of the review, lines 369-370, it says that multidisciplinary board communication has been shown to improve survival by providing a more comprehensive review of possible options. How, and by what percentage, is not mentioned.

Author Response

  1. We have now indicated the source (PMID:32606456) for Figure 1.
  2. Per suggestion, we have changed the terminology from CCA to iCCA in the text to align with Table 1 (except for the approval sentences for FGFRi and IDH1i, which aligns with the approved indication).
  3. We have not made this change because Table 1 only includes genetic alterations that have been shown to be targeted with agents in clinical trials or FDA approved therapies. Additionally, the genetic alterations specified in the table were chosen to align with the most widely studied alterations and content within the NCCN guidelines.
  4. We have removed the sentence "Communication in multidisciplinary boards has been shown to improve survival by providing a more comprehensive review of possible options" since it not fully aligned with the specific discussion.

Round 2

Reviewer 3 Report

The author has given a reasonable explanation based on my previous questions